# Genome-Wide Identification of *CsATGs* in Tea Plant and the Involvement of *CsATG8e* in Nitrogen Utilization

**DOI:** 10.3390/ijms21197043

**Published:** 2020-09-24

**Authors:** Wei Huang, Dan-Ni Ma, Hong-Ling Liu, Jie Luo, Pu Wang, Ming-Le Wang, Fei Guo, Yu Wang, Hua Zhao, De-Jiang Ni

**Affiliations:** 1Key Laboratory of Horticultural Plant Biology of Ministry of Education, Huazhong Agricultural University, Wuhan 430070, China; huangwtea@webmail.hzau.edu.cn (W.H.); madanni@webmail.hzau.edu.cn (D.-N.M.); liuhongling@webmail.hzau.edu.cn (H.-L.L.); luojie@mail.hzau.edu.cn (J.L.); pwang@mail.hzau.edu.cn (P.W.); wangmingle@mail.hzau.edu.cn (M.-L.W.); guofei@mail.hzau.edu.cn (F.G.); catea37@mail.hzau.edu.cn (Y.W.); nidj@mail.hzau.edu.cn (D.-J.N.); 2College of Horticulture and Forestry Sciences, Huazhong Agricultural University, Wuhan 430070, China; 3Hubei Engineering Technology Research Center for Forestry Information, Huazhong Agricultural University, Wuhan 430070, China

**Keywords:** autophagy-related genes, *Camellia sinensis*, *CsATG8e*, nitrogen

## Abstract

Nitrogen (N) is a macroelement with an indispensable role in the growth and development of plants, and tea plant (*Camellia sinensis*) is an evergreen perennial woody species with young shoots for harvest. During senescence or upon N stress, autophagy has been shown to be induced in leaves, involving a variety of autophagy-related genes (*ATGs*), which have not been characterized in tea plant yet. In this study, a genome-wide survey in tea plant genome identified a total of 80 *Camellia Sinensis* autophagy-related genes, *CsATGs.* The expression of *CsATG8s* in the tea plant showed an obvious increase from S1 (stage 1) to S4 (stage 4), especially for *CsATG8e*. The expression levels of *AtATGs* (*Arabidopsis thaliana*) and genes involved in N transport and assimilation were greatly improved in *CsATG8e-*overexpressed *Arabidopsis*. Compared with wild type, the overexpression plants showed earlier bolting, an increase in amino N content, as well as a decrease in biomass and the levels of N, phosphorus and potassium. However, the N level was found significantly higher in APER (aerial part excluding rosette) in the overexpression plants relative to wild type. All these results demonstrated a convincing function of *CsATG8e* in N remobilization and plant development, indicating *CsATG8e* as a potential gene for modifying plant nutrient utilization.

## 1. Introduction

Tea plant (*Camellia sinensis*, (L.) O. Kuntze), a widely cultivated horticultural crop, is a perennial evergreen woody plant [1]. Nitrogen (N) is one of the most important contributors for tea plant growth and leaf quality due to the harvest of leaves several times a year [2]. A large amount of N is required for rounds of buds sprout for a whole year to achieve considerable yield and top quality, which is especially true in spring. Additionally, pruning is usually performed by the end of plucking season, leading to the removal of a significant amount of N. All this suggests a huge demand of N in tea plantation. To solve this demand, excessive N is usually fertilized to maintain the vigorous growth and tea yield, resulting in many environmental problems [3,4]. Meanwhile, plants also take N through leaves [5,6], but no such genes have been reported in tea plant yet. Apart from the uptake of N from soil as well as foliar, N remobilization within plants plays an important role in meeting the demand of young leaves. Under insufficient N conditions, N remobilization becomes a more important N source for growing shoots [7].

Autophagy (macro-autophagy) is a highly conserved cellular degradation process, with portions of cytosol and organelles broken down and the resulting macromolecules eventually recycled [8]. During this process, the unwanted macromolecular substances or damaged organelles are first sequestered into a double membrane structure called an autophagosome, followed by fusing the outer membrane of the autophagosome with the vacuolar membrane to release the cargos and inner membrane structures into the vacuole for degradation and recycling [9]. Under normal growth conditions, autophagy occurs at a low basal level to maintain cell homeostasis. However, in response to stresses of nutrient-limited conditions and senescence, autophagy will be stimulated to alleviate these stresses for plant survival [10,11]. In plants, autophagy has been demonstrated to associate with N remobilization during leaf senescence and grain filling [12,13]. Several confirmation studies have been performed on autophagy mutants of *Arabidopsis* and cereal crops. For example, in rice, Wada et al. [14] showed that the N remobilization in senescent leaves was suppressed in the mutant *Osatg7-1* (*Oryza sativa*, Os). In maize, the investigation of N partitioning in the mutant *Zmatg12* (*Zea mays*, Zm) showed an impaired N remobilization, leading to a significant decrease in seed yield [11]. In *Arabidopsis*, under both low and high nitrate conditions, supplying ^15^NO_3_^−^ during the vegetative stage led to a sharp decrease in ^15^N remobilization in the mutants *atg5-1* and *atg9-2* relative to wild type in seeds at harvest [15].

Autophagy (ATG) genes, the main implementers and regulators of the autophagy process, were firstly characterized in yeast [16]. Since then, a number of *ATG* genes in single copy or subfamilies have been gradually identified in plants, such as *ATG1-13*, *16*, *18*, *101*, *VPS15* and *VPS34* [8,17,18]. According to the process of autophagy, these genes were classified into six functional complexes, with three of them as core autophagy genes in yeast, mammals and plants, including ATG9 recycling system, PI3K (Phosphoinositide 3-kinase) nucleation complex and two ubiquitin-like conjugation systems [18]. Genome-wide ATG genes have been systemically identified in many higher plants, such as 40 *AtATGs* in *Arabidopsis* [19] (*Arabidopsis*, At), 33 *OsATGs* in rice [14], 45 *ZmATGs* in maize [11] and 35 *VvATGs* in grapevine [20] (*Vitis vinifera*, Vv). However, no systemic reports are available about *ATG* genes in tea plant yet. In 2017, the first genome data for tea plant were released [21], followed by the successive release of several other genome data of *Camellia sinensis* for different cultivars [22,23], which provide useful information for understanding the *ATG* gene families in *Camellia sinensis*.

ATG8, a ubiquitin-like protein conjugated to phosphatidylethanolamine on the autophagic membrane, plays a central role in autophagy [24]. In plants, *ATG8* is a family with multiple numbers, such as nine *AtATG8s* in *Arabidopsis* [25], six *VvATG8s* in grapevine [20] and five *OsATG8s* in rice [14]. Studies have indicated that *ATG8* is beneficial to the plant performance in N utilization. For example, in apple, the expression of *MdATG8i* was induced in response to leaf senescence and N depletion, and the heterologous expression of *MdATG8i* (*Malus domestica*, Md) in *Arabidopsis* enhanced vegetative growth, leaf senescence and tolerance to N limitation [26]. Similarly, in rice, the expression of *OsATG8c* was enhanced during N starvation conditions, and the *35S-OsATG8c* rice plants showed an increase in yield as well as nitrogen uptake efficiency (NUpE) and nitrogen use efficiency (NUE) under both optimal and suboptimal N conditions [27]. Recently, Chen et al. [28] have shown that the overexpression of *AtATG8s* (*AtATG8a*, *AtATG8e*, *AtATG8f and AtATG8g*) with different promoters enhanced autophagosome number, promoted autophagy activity and increased the N remobilization efficiency under optimal N conditions without a negative effect on yield or biomass. Besides N, autophagy is also involved in recycling other nutrients, such as iron, zinc, manganese and sulfur [29,30,31]. Despite the characterization of the involvement of *ATG8s* in N remobilization and the participation of autophagy in recycling other nutrients in a number of plant species, little is known about them in tea plant.

In this study, a comprehensive investigation was performed about the *CsATG* families in the tea plant genome and a total of 80 *CsATGs* were identified. The gene structures, conserved domains and localizations on chromosomes were analyzed. Additionally, the phylogenetic trees were constructed along with their homologous genes from *Arabidopsis*, grapevine and sweet orange. Furthermore, we examined the expression patterns of *CsATG8s* in the leaves collected at four different stages from three tea plant cultivars, and the importance of autophagy in tea plant was explored by analyzing the expression profiles of 16 *CsATGs* in response to various N conditions. Finally, *CsATG8e* was functionally characterized to identify its functional mechanisms in response to various N supplies by heterologous expression in *Arabidopsis*.

## 2. Results

### 2.1. Genome-Wide Identification of CsATGs in Tea Plant

A BLASTp (Protein Basic Local Alignment Search Tool) search was performed using ATG sequences from *Arabidopsis*, *Citrus sinensis* and *Vitis vinifera* as the queries against the tea plant genome database. After confirming the conserved domains of ATGs in the predicted sequences, a total of 80 putative *CsATGs* from 24 subfamilies were identified. Among them, the subfamilies vary greatly in the number of gene members (Appendix A), with the largest number in the *CsATG18* subfamily (18 members), followed by the *CsATG8* subfamily (12 members). Another twelve of the 24 subfamilies consist of more than one member, with 8 members in the *CsATG16* family, 5 members in the *CsATG1* family, 4 members in both *CsATG20* and *CsVPS35* families and the remaining ten subfamilies contain a single member, including *CsATG2*, *CsATG5*, *CsATG6*, *CsATG10*, *CsATG11*, *CsATG13*, *CsATG101*, *CsATI*, *CsNBR1* and *CsVPS34*. The members in some subfamilies were named based on the similarity to the other three species, while the other members were named randomly, as shown in Appendix A.

The physiological and biochemical parameters of all family members identified from tea plant are shown in Appendix A. The genomic sequence length of the 80 genes ranges from 375 bp (*CsATG1b*) to 39,472 bp (*CsATG11*) with open reading frame (ORF) length varying between 219 bp (*CsATG12b*) and 5994 bp (*CsATG2*). The protein size of the 80 proteins ranges from 72 aa (*CsATG12b*) to 1997 aa (*CsATG2*), with an average molecular weight (MW) of 8.2 to 218.7 kDa. The predicted theoretical isoelectric point (PI) ranges from 4.5 (*CsATI*) to 9.9 (*CsATG18a*), with 40 proteins under 7.0, 38 proteins above 7.0 and 2 at 7.0, indicating that the ATG proteins are widely distributed under alkaline, acidic and neutral conditions. Collectively, the 80 *CsATGs* vary greatly in the length of genome and ORF, protein size, MW and PI, even in a subfamily of *ATG8* or *ATG18*, implying their functional divergence.

### 2.2. Bioinformatic Characterization of the 80 CsATGs

#### 2.2.1. Chromosomal Distribution of *CsATGs*

The chromosomal locations of *CsATGs* are shown in Figure 1a. Among the 80 *CsATGs*, 63 are widely distributed across the 15 chromosomes, and the other 17 genes are located on the 17 unassembled scaffolds (or assembled contigs), due to the incomplete physical map of *Camellia sinensis*. Chromosome 1 was predicted to contain the largest number of members (nine), followed by chromosome 2 (eight) and chromosome 4 (seven). On chromosomes 5, 9, 12, 13, 14 and 15, each contained two *ATG* members. As duplication usually contributes to the expansion of gene families, we investigated the duplication patterns of genes in one of the subfamilies containing more than one member. No tandem duplication gene was found, while a total of 31 genes were produced from the eight subfamilies of *CsATG1*, *CsATG3*, *CsATG7*, *CsATG8*, *CsATG16*, *CsATG18*, *CsVPS15* and *CsVPS35* by whole genome duplication (WGD), which constructed 25 pairs of collinear genes (Figure 1b, Appendix A). This suggests that WGD plays an important role in duplication of these gene subfamilies in tea plant. We also investigated the evolutionary processes in subfamily by the Ka/Ks (the ratio of nonsynonymous substitution to synonymous substitution) value but failed to obtain the value for one pair of genes (*CsATG8i* and *CsATG8h*) because they share the same coding sequence (CDS) despite their different localizations on chromosomes. Meanwhile, the Ka/Ks value is slightly higher than “1” (1.2279) for another pair of genes (*CsATG18q* and *CsATG18r*), but lower than “1” for the other 23 pairs of collinear genes. Purifying selection usually works with Ka/Ks < 1; otherwise, positive selection occurs once the value is far higher than “1” [32]. Overall, purifying selection plays a primary role in the *ATG* subfamily members and positive selection is also necessary during the evolutionary history.

#### 2.2.2. Phylogenetic Analysis

To explore the phylogenetic relationship among proteins in a subfamily, the neighbor-joining phylogenetic trees of the 24 subfamilies were constructed respectively based on the predicted amino acid sequences of each ATG subfamily in tea plant along with the ATG proteins from *Arabidopsis* and the two woody plants of grapevine (*Vitis vinifera*) and sweet orange (*Citrus sinensis*) (Figure 2). For the ATGs with a single member in a subfamily, the CsATGs were closely clustered with their homologues in a grapevine, excluding CsNBR1, which was more closely clustered with sweet orange than the other two species. As for the subfamilies with multiple members, some were clustered into one branch, such as CsATG3, CsATG4, CsATG14, CsATG20 and CsVPS15, and some were clustered into several branches, including CsATG1, CsATG8, CsATG9, CsATG12, CsATG16, CsATG18 and CsVPS35. Intriguingly, the four members in CsATG1 (CsATG1a, CsATG1c, CsATG1d and CsATG1e) were grouped more closely with the homologous genes in *Arabidopsis* than woody plants, indicating their potential functional similarity to *Arabidopsis*. These results suggested a closer relationship of tea plant with woody plants, especially grapevine, and also implied its potential functional differentiations within multiple subfamilies.

#### 2.2.3. Gene Structures and Conserved Domains

The gene structures of all ATGs were presented and the domain compositions were analyzed by comparison with the corresponding proteins of *Arabidopsis* (Appendix A). As shown in Figure 3, among these 80 *Cs**ATGs*, there was only one exon in the three genes of *CsATG16f*, *CsVTI12a* and *CsVTI12b*. However, for the other *Cs**ATGs*, the number of exons ranged from two (*CsATG1b*, *CsATI* and *CsATG16g*) to 25 (*CsVPS35b*). The variations in exon number can show the diversity in a subfamily with multiple members. For instance, no variation was found in the exon number in the subfamily of *CsATG3* (9 exons), *CsATG7* (14 exons), *CsATG14* (12 exons), *CsATG20* (10 exons) and *CsVPS15* (11 exons). A limited variation was observed in the number of exons in the subfamilies of *CsVPS35* (22 and 25 exons) and *CsATG8* (5 and 6 exons). A great variation was found in the number of exons for some subfamilies, ranging from 2 to 11 in *CsATG1*, 3 to 11 in *CsATG18* and 1 to 16 in *CsATG1*. For the untranslated regions (UTRs), the longest one was found in *CsATG18c* at 5′-UTR, with both 5′-UTR and 3′-UTR observed in 33 genes, only 5′-UTR in 6 genes and just 3′-UTR in 11 genes. Additionally, we analyzed the conserved domains by comparing the 80 ATG proteins with the corresponding proteins in *Arabidopsis* (Appendix A). Except for *CsATI* without the domain of *AtATI* in *Arabidopsis*, the remaining 79 genes had 1 to 3 (5 for *CsNBR1*) domains, which can be found in *Arabidopsis* for the same subfamily, including ATG_C, Chorein_N, Autophagy_N, Peptidase_C54, E1_like_apg7, Ubl_ATG8, WD40, BCAS3, etc. However, some differences were observed between *Arabidopsis* and tea plant. For example, in *ATG1*, the domain PKc_like superfamily was found in the four subfamily members (*CsATG1a*, *CsATG1b*, *CsATG1d* and *CsATG1e*) in tea plant, but not in *Arabidopsis*. Meanwhile, for the *ATG11* subfamily, an APG17 superfamily domain was found in *Arabidopsis*, but not in tea plant. The *CsATGs* in a subfamily with the same domains might show functional similarity, in contrast to functional differentiation between subfamily members with different domains.

### 2.3. Expression Profiles of CsATGs in Camellia sinensis

#### 2.3.1. Expression Patterns for *CsATG8s* in Tea Cultivars

For evergreen perennial plants, the nutrient allocation within the plant is assumed to vary dynamically within tissues at different stages. To illustrate this assumption, we investigated the expressions of *CsATG8s* (ten out of twelve members, with the same amino acid sequences in *CsATG8h* and *CsATG8i*, *CsATG8k* and *CsATG8l* respectively, Appendix A) in leaves at four different stages in the three green cultivars of *Fuding Dabai*, *Echa 10* and *Zhongcha 108*. As shown in Figure 4a, all the tested genes showed the highest expression at S4 except for *CsATG8d* in *Fuding Dabai* and *CsATG8k* in *Echa 10*. Additionally, an upward tendency was observed from S1 to S4 in all the three cultivars in the expression of *CsATG8a*, *CsATG8e* and *CsATG8g*, especially *CsATG8e*, suggesting the potential role of *CsATG8e* in nutrient redistribution in tea plant.

#### 2.3.2. Expression Patterns for *CsATGs* in Response to N

To explore the expression patterns of *CsATGs* in response to N, *Zhongcha 108* was selected and treated with two N levels: sufficient (5 mM N, 2.50 mM NH_4_NO_3_) and deficient (0.25 mM N, 0.125 mM NH_4_NO_3_), based on the expression patterns of *CsATG8s* in *Fuding Dabai*, *Echa 10* and *Zhongcha 108*. As the total number of identified *CsATGs* was up to 80, sixteen *CsATGs* were randomly chosen here from the members of relatively large subfamilies or key genes in autophagy process. qRT-PCR (quantitative reverse-transcription polymerase chain reaction) was used to investigate the expression patterns of the selected *CsATGs* in the young and the mature leaves under exposure to the two N conditions (5 and 0.25 mM). As shown in Figure 4b, a steady up-regulation was observed in both young and mature leaves in most of the sixteen *CsATGs* under N deficiency and sufficient conditions relative to their respective control, with the upregulation being more obvious under the sufficient N condition, especially for *CsATG18f* in both young and mature leaves. Additionally, the changes were more obvious in mature leaves than in young leaves when compared with their respective control. Among them, the transcript levels of *CsATG8e* showed the most obvious changes in both leaves under both sufficient and deficient N conditions. Overall, all the selected *CsATGs* were stimulated in response to N, with a higher variation in the response of *CsATG8e* to N changes in tea plant.

### 2.4. Cloning and Functional Characterization of CsATG8e

#### 2.4.1. Cloning of *CsATG8e* and Comparison with ATG8s in *Camellia sinensis* and *Arabidopsis*

To investigate its function in tea plant, the 360 bp ORF of *CsATG8e* was cloned from *Fuding Dabai*. *CsATG8e* shared 4.28% to 89.08% identity with the *AtATG8* members in *Arabidopsis*, with the highest sequence similarity to *AtATG8c*, while the smallest similarity to *AtATG8e* (Appendix A). Meanwhile, *CsATG8e* shared 45.38% (*CsATG8l*) to 95.80% (*CsATG8c*) identity with the members of the *CsATG8* subfamily in *Camellia sinensis* (Appendix A). These results illustrated a higher similarity with homology proteins in tea plant than in *Arabidopsis*, and *CsATG8e* showed more sequence similarity to *ATG8c* in both *Arabidopsis* and *Camellia sinensis*. Based on the amino acid sequences, ATG8 proteins can be divided into three groups: clade I (a), clade I (b) and clade II [33]. However, according to our phylogenetic analysis of CsATG8s (Figure 2), no clear classification was found for CsATG8s in tea plant. Thus, we investigated the amino acid sequences of CsATG8 proteins, and found a lack of an extra amino acid residue in CsATG8d and CsATG8f at the C-terminus after the glycine residue, similar to AtATG8h and AtATG8i, which belong to clade II. In our phylogenetic analysis, CsATG8e and AtATG8d (a member from clade I (a)) were classified into one group, implying CsATG8e belongs to clade I (a).

#### 2.4.2. Transgenic Plants Promote Development but Decrease Biomass

To gain insight into how *CsATG8e* functions in response to N, the *CsATG8e-*overexpressed *Arabidopsis* plants were generated, and hydroponic growth conditions were established to figure out the progressive changes in response to three N regimes: low (0.25 mM N, 0.125 mM NH_4_NO_3_), moderate (1 mM N, 0.50 mM NH_4_NO_3_) and sufficient (5 mM N, 2.50 mM NH_4_NO_3_). After a 20-day culture under the sufficient N condition, an obvious difference was observed in the vegetative shoot between wild-type (WT) plants and overexpression (OE) plants (Figure 5a). Specifically, the inflorescences were visible in the OE plants while the WT plants were still in the vegetative growth stage, indicating the significant effect of *CsATG8e* on the development of overexpression plants.

After two weeks of growth under the three N regimes, compared with WT, the rosette of OE plants showed obvious purple and serious senescence, and a smaller size, regardless of the N supply level (Figure 5b). We examined the biomass of roots (Root), rosettes (Rosette) and aerial part excluding rosette (APER), and a decrease was observed in the biomass of Root, Rosette and APER in OE plants versus WT plants (Figure 5c). Considering that the buds or the tender leaves are usually harvested in tea plant, while in *Arabidopsis*, the APER came up after rosette formation, we investigated the biomass allocation in aerial part and defined biomass allocation in APER as APER/(APER + Rosette) in this study. The biomass allocation in APER was found to be significantly higher than that of WT under all the three N regimes (Figure 5d). Collectively, the overexpression of *CsATG8e* in *Arabidopsis* led to earlier flowering and decreased biomass, regardless of N conditions.

#### 2.4.3. Transgenic Plants Enhance N Use Capacity under Both Sufficient and Deficient N Conditions

To confirm whether autophagy differences exist between wild-type and transgenic plants, we investigated the expression patterns of 11 genes (*AtATG9*, *AtNBR1* and 9 genes in ubiquitin-like conjugation pathways) in the leaves after N treatment at 0.25 and 5 mM for one week (Figure 6a,d). A significantly (*p* < 0.01) upregulated expression was observed in OE lines under both N treatments, except for *AtATG8**c* under the deficient N condition. We further examined the expression profiles of some genes related to N uptake, transport and assimilation. In root, a significant (*p* < 0.01) difference was observed in the expression patterns of the 5 tested nitrate and ammonium genes (*AtAMT1;1*, *AtAMT1;2*, *AtAMT1;3*, *AtNRT1;1* and *AtNRT2;2*) between deficient and sufficient N conditions (Figure 6b,e). Under low (0.25 mM) N treatment, a significant (*p* < 0.01) decrease was observed in OE lines versus wild-type except for *AtNRT1;1*, with a slight decrease, in contrast to a strongly upregulated expression in OE plants versus WT under sufficient (5 mM) N, except for a sharp decrease in the expression of *AtAMT1;3* and *AtNRT1;1* in OE plants. In leaves, both the amino acid transporters (*AtAAP1*, *AtAAP4*, *AtAAP5* and *AtAAP6*) and genes involved in N assimilation (the nitrate reductase encoding gene *AtNIA1*, the Gln synthetase encoding gene *AtGLN1;1* and Gln 2-oxoglutarate aminotransferase coding gene *AtGLU1*) were upregulated to different degrees under the two N conditions (Figure 6c,f). Specifically, under 0.25 mM N treatment, a strongly (*p* < 0.01) increased expression was observed in OE lines versus WT for all examined genes except for *AtAAP6*. Similarly, under 5 mM N treatment, a significantly (*p* < 0.01) elevated expression was found in OE lines for all the tested genes related to N transport and assimilation.

The aforementioned results indicate that the overexpression of *CsATG8e* in *Arabidopsis* could enhance the expression of autophagy-related genes, several amino acid transporters and genes related to N assimilation, regardless of the N condition. Meanwhile, the expressions of the genes related to N uptake all showed a decrease under low N stress, indicating the potential improvement of the N utilization capacity in transgenic plants. However, the mechanism for the improvement is different and depends on N conditions.

#### 2.4.4. Transgenic Plants Increase Amino N

To evaluate the potential effect of *CsATG8e* on N remobilization during autophagy, we measured and compared the amino N levels between OE lines and WT plants. In Figure 7, the level of amino N was shown to increase in OE plants versus WT plants under the three N conditions, with a strikingly elevated concentration observed under the 1 mM N condition. The results indicate that the *CsATG8e* OE lines outperform the WT plants in amino N content in all the three N regimes from deficiency to sufficiency.

#### 2.4.5. Transgenic Plants Improve N Allocation in APER

Furthermore, we investigated how *CsATG8e* affects the distribution and accumulation of N under different N conditions. In APER, rosettes and roots, the N content in OE plants was shown to decrease dramatically relative to WT (Figure 8a) but varied with N conditions in the investigated parts (Appendix A). Generally, the N concentration increased with increasing N supply for all parts in OE lines and WT plants. When compared to WT plants, OE plants showed a decrease in the N concentration of rosettes and roots to a different degree under all three N conditions. However, in APER, the downregulated concentration of N was only observed under the 5 mM N condition, despite a non-statistical significance (*p* > 0.05). Additionally, we investigated the N allocation in APER (Figure 8b). Interestingly, the N allocation in APER was significantly increased in OE plants versus WT plants, indicating that *CsATG8e* could improve N allocation in APER independently of the N condition.

#### 2.4.6. The Decrease in the Content of P and K in Transgenic Plants Is Mainly Attributed to Biomass Rather than the Concentration of P and K

Phosphorus (P) and potassium (K) are two other essential macronutrients in plants. To evaluate whether differences exist in the utilization efficiency of nutrients between OE lines and WT plants across the three tested N regimes, we determined the contents of P and K within plants (Appendix A). Under all three N conditions, the content of P and K showed a sharp decrease in OE lines versus WT in roots, rosettes and APER. A comparison between OE lines and WT (Appendix A) revealed no significant changes of P concentration in APER, while a significant increase in the rosettes of OE versus WT plants only under 1 mM N condition. In roots, the P concentration showed an increase in OE plants under all three N conditions, with a significantly (*p* < 0.05) higher P concentration observed at 1 and 5 mM N. As for K concentration, a similar downregulation was observed in the three parts of OE versus the WT plants under the 5 mM N condition (Appendix A). Under the 1 mM N condition, the K levels in OE plants showed a detectable (*p* > 0.05) increase in APER and roots and a decrease in rosettes when compared with WT.

Additionally, we calculated the P and K allocation in APER. Under the 1 mM N condition, the K allocation in APER showed a consistent increase in transgenic plants versus WT. These results suggest that it is the biomass rather than the concentration of P and K that contributes to the changes in the P and K allocation within the *CsATG8e* OE plants.

## 3. Discussion

*ATG* genes have been widely reported to participate in the autophagy process, an evolutionarily conserved intracellular process for balancing protein synthesis and degradation [34], and a number of ATGs have been identified in plants [8]. For example, in the model plant *Arabidopsis*, more than 40 *ATG* genes have been identified [18,25], and in other plants, 45 genes were identified in maize (*Zea mays*) [11], 33 in rice (*Oryza sativa*) [35], 30 in tobacco (*Nicotiana tabacum*) [36], 35 in grapevine (*Vitis vinifera*) [20] and 35 in sweet orange (*Citrus sinensis*) [37]. However, no *ATG* genes have been reported to be identified in tea plant yet. In the present study, a total of 80 *ATG* genes were identified in tea plant, a number obviously larger than that in other plants, nearly twice the number in *Arabidopsis*, which can be attributed to the following three reasons. Firstly, 24 subfamilies were found in the tea plant genomic database, which is more than the number in other plant species, such as 20 subfamilies in maize, 13 in rice, 16 in tobacco, 21 in grapevine and 19 in sweet orange [11,20,35,36,37]. Secondly, different methods were used to search for the homologues. In previous reports, BLASTN search or keyword “autophagy” was applied, and in most cases, the sequences from *Arabidopsis* or/and rice were the only query-sequences [20,36,37]. In the present study, the query-sequences of *Arabidopsis* and another two woody plants were used to search for similar sequences. Finally, the improvement of genomic data for tea plant is still ongoing to obtain high-quality genomic data.

The identification of a large number of *ATG* members in tea plant implies the occurrence of duplication events of the ATG family, which was especially obvious for the subfamilies of *CsATG18*, *CsATG8* and *CsATG16*, whose total number occupies nearly half of all the identified *CsATG* members. Additionally, some of the *CsATG* genes share high similarity in a subfamily. For example, the two members (*CsATG8h* and *CsATG8i*) shared the same CDS with different gene structures and locations on chromosome (*CsATG8h* on chromosome 3 and *CsATG8i* on contig 424) (Figure 1a,b and Figure 3). For these genes, functional redundancy might be more frequently associated with their high similarity.

Unlike in yeast, the multiple members of a subfamily were usually observed for *ATG* genes in plants [17,38]. In our study, ten *ATG* genes were identified as a single copy in a separate subfamily. Among them, *ATG2* and *ATG5* are widely present as a single copy in higher plants, other than the genomic data we have mentioned above, which has also been proven in banana [39], pepper [40] and foxtail millet [41]. *ATG4* and *ATG9* are usually present as a single copy gene in many plants, such as tobacco [36], sweet orange [37], grapevine [20] and pepper [40], but with multiple members in a subfamily in rice [35]. In the present study, these genes were in small subfamilies with three and two members in *ATG4* and *ATG9,* respectively. Moreover, in higher plants, the subfamilies containing multiple members are always found in *ATG1*, *ATG8* and *ATG18* [8]. A similar result was found in tea plant for these subfamilies in the present study. Furthermore, we found that the members in a subfamily tend to share the same domains (Figure 3). In the subfamily of *CsATG18*, all members have the WD40 domain and seven members share an extra BCAS3 domain behind the WD40 domain, while in the *CsATG8* subfamily, ten of the members contain the Ubl_ATG8 domain and the other two have the Atg8 domain instead. Considering a closer distance in the phylogenetic trees (Figure 2), these results illustrated a closer relationship among the members with the similar conserved domains. Generally, the conserved domains and a relatively closer distance with woody species in the phylogenetic trees indicate a similarity in their functions, while separate branches and some tightly clustered members in tea plant suggest that some genes might be either in neofunctionalization or in redundancy.

In the species of *Arabidopsis* and rice, the functions of ATG8 subfamily members are shown to be involved in the utilization of nutrients, especially N [27,28,42]. Previous studies have confirmed that autophagy occurs at a very low level under normal conditions, while it is stimulated during senescence or in stressful conditions [43,44]. In this study, the expressions of 10 members in *CsATG8s* from the three tea cultivars mostly showed the highest expression at S4, while the lowest at S1 (Figure 4a), where the S4 leaves represented ageing leaves, while S1 tissues were young developing leaves with the requirement of a significant amount of nutrient for their growth and development, which can serve as supplementary to the previous results. Because of the central role of *ATG8* in autophagy, the expression patterns of *CsATG8s* in the four types of leaves from the three tea cultivars indicated that autophagy plays an important role in tea plant, implying the substrate remobilization from S4 to S1. However, the non-uniform expression patterns of the 10 *CsATG8s* indicated that they may have different functions or work at different occasions in tea plants. Additionally, the obvious upward tendency in the expression of *CsATG8e* from S1 to S4 suggested a specific function of this gene in nutrient redistribution in tea plant.

N has been demonstrated as one of the efficient contributors to stimulate autophagy [28,45]. In our study, we demonstrated that the expressions of sixteen genes were highly upregulated by N stimuli, especially under the sufficient N condition (5 mM N). However, previous studies have shown that autophagy can be more easily induced by nutrient starvation [46]. This discrepancy in results might be due to the different states of plants before N treatment in previous studies, where the plants were cultured in the sufficient N condition, while in our experiment, plants were subjected to ten days of N starvation. Additionally, the increased expression levels under both deficient and sufficient N conditions, especially in mature leaves, indicated that the autophagy process plays an important role in N economy within tea plant. Furthermore, the transcript levels of *CsATG8e* were found to change greatly within treatments, indicating an intimate connection of *CsATG8e* with N in tea plant.

The important role of autophagy in nutrient recycling has been proved in *Arabidopsis* [47,48]. In crops like rice [42,49,50] and soybean [51], a number of members in the ATG8 subfamily have been shown to participate in N utilization. Previous studies have shown that the ATG9 complex and ubiquitin-like conjugation pathways are indispensable in the process of autophagy and NBR1 is a typical receptor for selective autophagy [25,33]. In this study, the expressions of *AtATG9*, *AtNBR1* and 9 genes in ubiquitin-like conjugation pathways were found to be upregulated in OE lines versus WT (Figure 6a,d), indicating the enhanced autophagy in *CsATG8e* transgenic plants. As reported in earlier studies, *AtNRT1;1* and *AtNRT2;2* showed low and high affinity for nitrate uptake respectively, and *AtAMT1;1*, *AtAMT1;2* and *AtAMT1;3* absorbed up to 95% of ammonium in *Arabidopsis*, which are the major genes related to N uptake in *Arabidopsis* [52]. In this study, the expression of most of these genes in OE plants were downregulated under the 0.25 mM N condition and upregulated under the 5 mM N condition compared with WT (Figure 6b,e), indicating the occurrence of a different mechanism for *CsATG8e-*overexpressed plants under different N levels. *AAP1*, *AAP4*, *AAP5* and *AAP6* have been confirmed to play important roles in the transport of amino acids, while *AtNIA1*, *AtGLN1;1* and *AtGLU1* are members involved in N assimilation [53,54]. In our study, the upregulated expression of these genes in transgenic plants versus control plants (Figure 6c,f) demonstrated the enhancement of N utilization in *CsATG8e-*overexpressed plants.

Moreover, we demonstrated that compared with WT, the OE lines showed an earlier reproductive stage and senescence, a decrease in biomass and the content of N, P and K, while an increase in amino N. The increase of amino acid level has been shown to result from proteolysis for reuse in plants via autophagy [9]. Clearly, the higher amino N level in OE lines can be attributed to the degradation of severely senescent leaves. The earlier productive stage has been reported in soybean [55] and rice [42]. However, the results of biomass and nutrient content in the present study were inconsistent with those of *SiATG8a* [41], *OsATG8a* [50] and *MdATG8i* [26], in which the transgenic plants showed better growth and accumulated more N, whereas a recent study has demonstrated that the overexpression of *ATG8* in *Arabidopsis* has no effect on vegetative biomass or plant development [28]. In general, the tea plant is a woody plant with new vegetative organs rather than reproductive organs as economic products. Here, we investigated the nutrient allocation in the APER of *Arabidopsis* before the plants entered the mature reproductive stage, and the OE plants were shown to have an increase over the WT plants in the allocation of biomass and the content of N, P and K, with a slight difference between them in the concentration of N, P and K, indicating that the nutrient content mainly comes from biomass. These findings suggest that *CsATG8e* performs a potential function in the allocation of nutrients or the development of plants.

## 4. Materials and Methods

### 4.1. Identification of CsATG Genes

To comprehensively identify the *ATG* superfamily members in tea plant, the genomes of *Arabidopsis thabidopsis* (https://www.Arabidopsis.org/) and *Camellia sinensis* in CSS_ChrLev_20200506 version (http://tpia.teaplant.org/index.html) [23] were first compared by BLASTpat the whole genomic level under E-values above 1 × 10^−5^. Next, 49 autophagy-related gene proteins (ATGs) from *Arabidopsis thabidopsis* were used to acquire the corresponding proteins in *Camellia sinensis*. Simultaneously, 37 and 35 autophagy-related protein sequences from *Citrus sinensis* (http://citrus.hzau.edu.cn/orange/) and *Vitis vinifera* (http://plants.ensembl.org/index.html) were used in a BLASTp search for the similar sequences in *Camellia sinensis* genome under E-values above 1 × 10^−5^. Based on the sequences from the above three BLASTp searches, another BLASTp search was performed in NCBI (National Center for Biotechnology Information, https://blast.ncbi.nlm.nih.gov/Blast.cgi) to ensure that the sequences were annotated as autophagy-related genes, and the conserved domains were further validated online in the NCBI Conserved Domain Database (https://www.ncbi.nlm.nih.gov/cdd/) according to the domains from *Arabidopsis* (Appendix A). The isoelectric point (PI) and molecular weight (MW) were calculated by the ExPASy website (https://web.expasy.org/compute_pi/). TBtools [56] was used to acquire the sequences from tea genome and analyze the length of genes.

### 4.2. Phylogenetic Tree Construction and Analysis of Conserved Motif and Gene Structure of CsATGs

To investigate the evolutionary relationship of the *CsATGs* and homologues with those of other plant species, neighbor-joining phylogenetic trees were constructed respectively based on the subfamily of CsATG proteins together with three other species, *Arabidopsis thaliana* (At) from TAIR (The Arabidopsis Information Resource, https://www.Arabidopsis.org/), *Vitis vinifera* (Vv) from Ensemble Plants (http://plants.ensembl.org/index.html) and *Citrus sinensis* (Cis) from the sweet orange genome database (http://citrus.hzau.edu.cn/orange/), with their corresponding Gene Bank accession numbers listed in Appendix A. MEGA-X (http://www.megasoftware.net) was first used to align the amino acid sequences and generate the unrooted phylogenetic tree by the neighbor-joining method with 1000 repetitions. The online tool iTOL (https://itol.embl.de/) was used to visualize the phylogenetic trees. The conserved motifs and domains were analyzed using MEME (Multiple Em for Motif Elicitation, http://alternate.meme-suite.org/tools/meme) and NCBI-CDD (Conserved Domain Datebase, https://www.ncbi.nlm.nih.gov/Structure/bwrpsb/bwrpsb.cgi) and visualized with TBtools. The exon-intron structures of the *CsATGs* were determined by comparing the coding sequences (CDS) with the corresponding genomic sequences using Gene Structure Display Server (GSDS, https://www.ncbi.nlm.nih.gov/cdd).

### 4.3. Plant Materials and Nitrogen Treatments

To examine the expression of *CsATG8s* in tea leaves at different development stages, leaves at four different growing phases, including one bud with two leaves or the same tenderness with two leaves near the bud that stops growing (Stage 1, S1), leaves attached to the red and green stems (Stage 2, S2), leaves attached to the gray stems (Stage 3, S3) and ageing leaves with a tendency to become yellow (Stage 4, S4), were collected on 28 September 2017 from tea plants of three different green tea cultivars in the Tea Plantation of HuaZhong Agricultural University (Wuhan, Hubei, China). The tea cultivars included *Fuding Dabai* and *Echa 10*, two national cultivars largely grown in China, and *Zhongcha 108*, a provincial cultivar mainly grown in Hubei province. The leaves with a similar and vigorous growth status were used in the experiments. All samples were collected and frozen immediately in liquid nitrogen and stored at −80 °C for RNA extraction.

To reveal the response of ATGs to different N regimes, *Zhongcha 108* was used to examine the expression profiles. The two-year-old cutting seedlings were planted in the plastic basin (48 plants per basin) containing 7 L of hydroponic culture in a chamber (22 °C/18 °C, 16 h light/8 h dark) for 15 days. The concentration of the nutrient solution was refreshed every five days, stepwise supplied at 1/8 to 1/4 and then 1/2 strength. Finally, the plants were cultured with full-strength solution for another two months until the new roots developed. For the N treatment, plants were first cultured in the solution without N for ten days, followed by exposure to the solution containing N at 0.25 mM (deficient, 0.125 mM NH_4_NO_3_), or 5.0 mM (sufficient, 2.50 mM NH_4_NO_3_) for 8 h. The samples of young leaves and mature leaves were collected and frozen immediately in −80 °C for RNA extraction, and the samples collected before the 8 h treatment were used as the control (CK). Here, the full-strength solution was based on Wan et al. [57] with slight changes, which contained 2.50 mM NH_4_NO_3_, 1 mM KH_2_PO_4_, 0.15 mM K_2_SO_4_, 0.44 mM CaCl_2_·2H_2_O, 0.43 mM MgSO_4_·7H_2_O, 0.125 mM Al_2_(SO_4_)_3_·18H_2_O, 2.1 µM Na_2_EDTA, 2.1 µM FeSO_4_·7H_2_O, 3.33 µM H_3_BO_3_, 0.50 µM MnSO_4_·H_2_O,0.51 µM ZnSO_4_·7H_2_O, 0.13 µM CuSO_4_·5H_2_O, and 0.17 µM Na_2_MoO_4_·2H_2_O, with pH 5.83 for the solution.

OE *Arabidopsis* plants were cultured in the soil in a growth chamber (22 °C/18 °C, 16 h light/8 h dark) to obtain homozygous T3 seeds. To investigate the N patterns mediated by *CsATG8e*, wild type (WT, Col-0) and two homozygous T3 lines with relatively higher expression levels were used to compare their growth, N level and allocation in response to the three different N regimes. Firstly, the seeds were surface-sterilized and sowed on plates containing 1/2 MS (Murashige and Skoog) medium (with 1% agar and 1% sucrose (w:v), pH = 5.83), then stratified at 4 °C for two days in the dark, and finally grown in a growth chamber (22 ± 1 °C and 16 h light/8 h dark) for two weeks. Next, the plants were transferred to fresh 1/2 modified Hoagland’s solution for 5 days, followed by culture for another 15 days in the full Hoagland’s solution, which contained 2.50 mM NH_4_NO_3_, 1.6 mM CaCl_2_·2H_2_O, 0.4 mM K_2_HPO_4_·3H_2_O, 0.7 mM K_2_SO_4_, 1 mM MgSO_4_·7H_2_O, 25 µM FeSO_4_·7H_2_O, 25 µM Na_2_EDTA, 9 µM MnSO_4_·H_2_O, 47 µM H_3_BO_3_, 0.8 µM ZnSO_4_·7H_2_O, 0.3 µM CuSO_4_·5H_2_O and 0.1 µM Na_2_MoO_4_·2H_2_O, with the full solution renewed every 5 days. Subsequently, the N in the solutions was replaced with low (0.25 mM N, 0.125 mM NH_4_NO_3_), moderate (1 mM N, 0.50 mM NH_4_NO_3_) and high (5 mM N, 2.50 mM NH_4_NO_3_) levels separately for another 14 days of culture, with each of the solutions renewed once a week until the samples were collected. Twenty days after transferring from plates, the grown plants were recorded for the first time and then treated with the three different N regimes as described above. One week after such treatments, samples of roots and leaves under low and high conditions were collected from at least three plants for gene expression analysis. Finally, at the end of the cultivation, the plants were photographed again and then sampled by separation into Root, Rosette and APER for measurement of biomass and N, P and K concentration.

### 4.4. RNA Extraction and Gene Expression Analysis

Total RNA was extracted from tea leaves, *Arabidopsis* leaves and respective roots using the Quick RNA Isolation Kit according to the manufacture’s protocol (Huayueyang, Beijing, China). The RNA was treated with gDNA Eraser to remove DNA contaminants (Aidlab, Beijing, China). Reverse transcription was performed using TRUEscript RT Kit (Aidlab, Beijing, China) with 1 μg RNA. The resulting cDNAs were diluted at 1:10 with ddH_2_O, and qRT-PCR was performed using an ABI StepOnePlus Real-Time PCR System (Applied Biosystems) with SYBR Green qPCR Mix (Aidlab, Beijing, China) following a recommended protocol. Fold changes in gene expression were calculated by comparing the C_T_ values using the 2^−ΔΔ^ method [58] with the internal control genes of *CsGAPDH* for *Camellia sinensis* and *AtGAPDH* for *Arabidopsis*. Three biological replicates were performed, and each replicate was collected from at least two independently grown sets of plants. Accession numbers and primer sequences are shown in Appendix A.

### 4.5. Cloning of CsATG8e and Generation of Transgenic Arabidopsis thaliana Plants

The ORF of *CsATG8* was firstly amplified with non-restriction enzyme primers from cDNA of *Fudingdabai* cultivar and cloned into pTOPO vector following the manufacturer’s instructions. After sequencing and alignment against the reference genome, the confirmed coding sequences were separately amplified with primers containing *Xba* l/*Xho* l and finally inserted into pBin35SRed [59]. The resulting constructs were introduced into *Agrobacterium tumefaciens* stain GV3101. Finally, transformation of *Arabidopsis thaliana* wild type was performed using the floral dip method [60]. Two independently homozygous T3 lines with *CsATG8e* at the high expression level were selected for further analysis. All primers used in vector construction are shown in Appendix A.

### 4.6. Determination of N, P and K Concentration and Amino N in Arabidopsis

Samples of Root, Rosette and APER were oven-dried at 120 ℃ for 10 min, followed by 75 ℃ for 5 days to a constant weight. Biomass was calculated as the sum of dry weights from the separated parts. To measure the total N, P and K concentration, the dried samples were firstly ground into fine powder, followed by digestion with H_2_SO_4_–H_2_O_2_. The concentrations of N and P were determined simultaneously using a flow injection analysis instrument (FIAstar 5000 analyzer; FOSS, Hilleroed, Denmark). The concentration of K was determined with a flame spectrophotometer (Sherwood M410).

To measure the amount of amino N, fresh samples (~0.15 g) of Root, Rosette and APER were weighed separately, followed by measuring the amount of amino N with the Ninhydrin Colorimetric Analysis method [61].

### 4.7. Statistical Analysis

For each determination, at least three independent replicates were carried out. Data were presented as mean ± standard deviation. OE lines and WT were statistically analyzed by Student’s *t*-test at significance levels of * *p* < 0.05 and ** *p* < 0.01 using the SPSS 21.0 software (IBM, Chicago, IL, USA). Significant differences in tea plant were indicated with lowercase letters using analysis of variance (ANOVA) Duncan’s test at *p* < 0.05 or *p* < 0.01. All the figures were drawn using the software of OriginPro 9.1 (OriginLab, Northampton, MA, USA) or TBtools and Adobe illustrator CC2019 (ADOBE, San Jose, CA, USA).

## Figures and Tables

**Figure 1 ijms-21-07043-f001:**
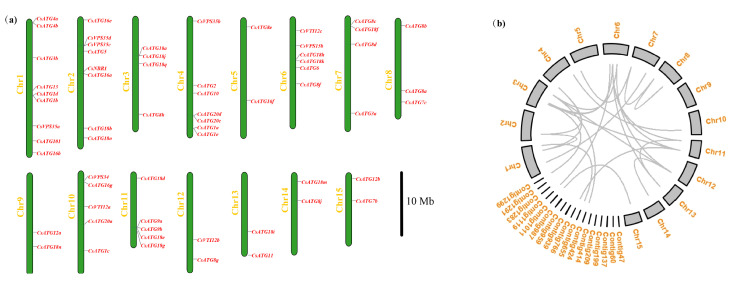
Chromosomal distribution and gene duplication of *CsATG* genes. (**a**) The distribution of 63 *CsATG* genes in 15 chromosomes. (**b**) The chromosomes and contigs containing the 80 *CsATG* genes, and the collinear pairs illustrated with one gray curve for one pair.

**Figure 2 ijms-21-07043-f002:**
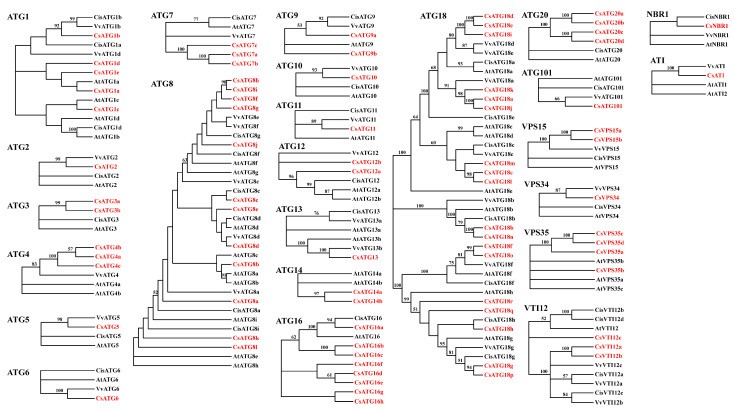
Phylogenetic analysis of *Camellia sinensis* and other plant ATG proteins. The neighbor-joining trees were constructed with 80 CsATGs of *Camellia sinensis* (Cs), 49 AtATGs of *Arabidopsis thaliana* (At), 37 CisATGs of *Citrus sinensis* (Cis) and 35 VvATGs of *Vitis vinifera* (Vv) using the MEGA-X software, with 1000 bootstrap replicates, and displayed without the tree root using the online tool iTOL (https://itol.embl.de/). The genes of tea plant are marked in red color. The bootstrap values higher than 50% are shown in the phylogenetic trees.

**Figure 3 ijms-21-07043-f003:**
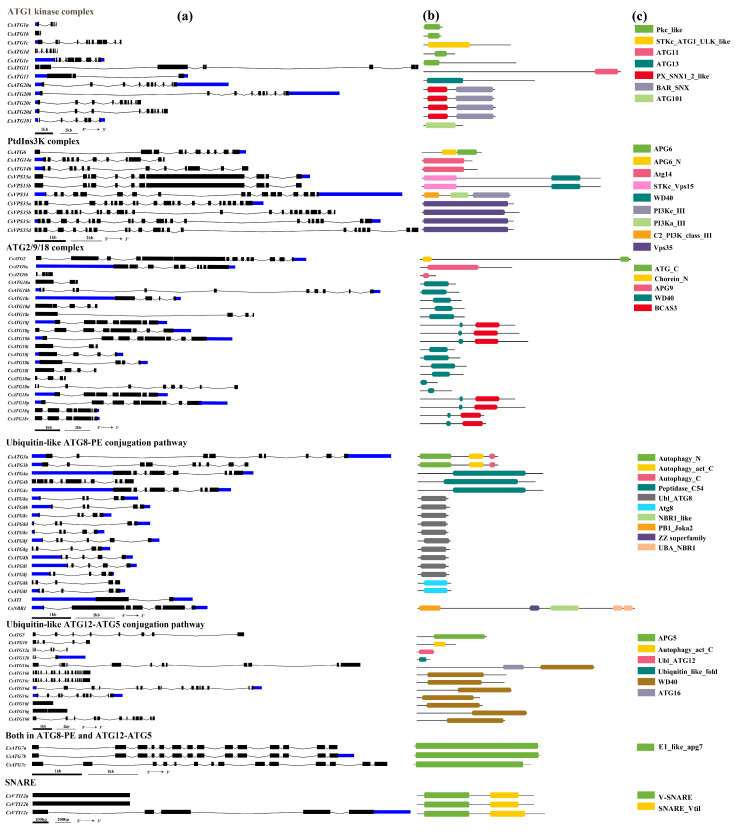
Gene structures and domains of *CsATG* genes in tea plant. (**a**) The exon-intron structures, with untranslated regions (UTRs) in blue rectangles, exons in black rectangles and introns in thin black lines. Domains of the corresponding genes on the left side (**b**,**c**). The 80 *CsATG* genes were presented according to their functions in autophagy.

**Figure 4 ijms-21-07043-f004:**
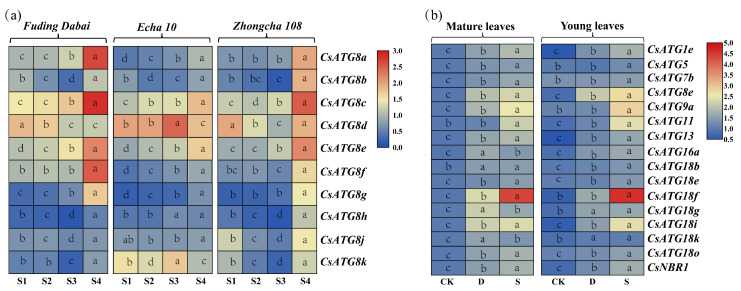
Expression patterns of *CsATG* genes in *Camellia sinensis*. (**a**) The expression of *CsATG8s* in the leaves at four different stages (S1, Stage 1; S2, Stage 2; S3, Stage 3; S4, Stage 4) in tea cultivars of *Fuding Dabai*, *Echa 10* and *Zhongcha 108*. (**b**) The expression of sixteen *CsATG* genes in *Zhongcha 108* under different N treatments. Y, young leaves; M, mature leaves; D, N-deficient, 0.125 mM NH_4_NO_3_; S, N-sufficient, 2.50 mM NH_4_NO_3_; CK, control. Different lowercase letters in rectangles indicate significant differences at *p* < 0.05 by analysis of variance (ANOVA) Duncan comparison.

**Figure 5 ijms-21-07043-f005:**
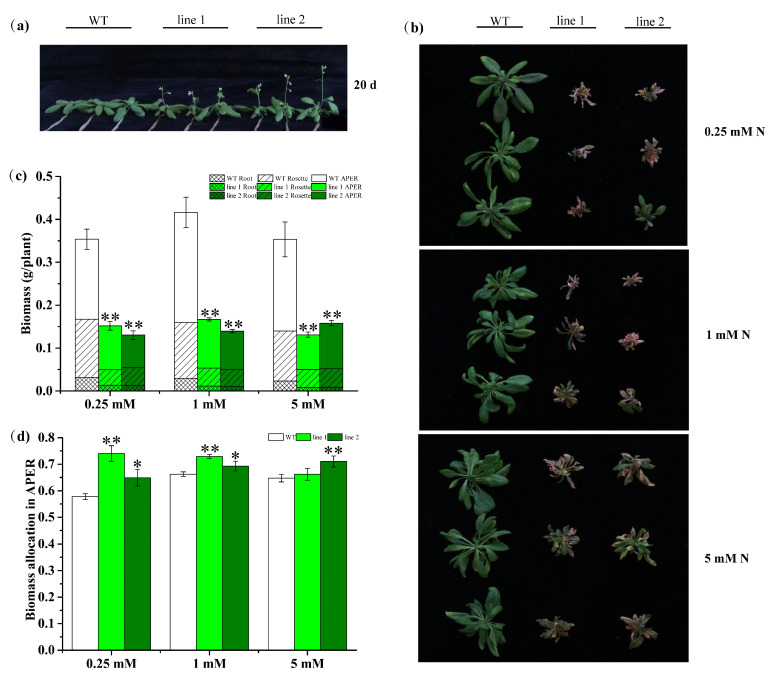
Comparison of growth performance between *CsATG8e-*overexpressed *Arabidopsis* (OE) and wild-type (WT) plants. (**a**) Growth of the WT and *CsATG8e* OE *Arabidopsis* after a 20-day culture at the sufficient N condition. (**b**) The rosette of OE and WT *Arabidopsis* after a 20-day culture under sufficient N, followed by another fourteen-day culture under different N levels. (**c**) The comparison of biomass in aerial part excluding rosette (APER), rosettes (Rosette) and roots (Root) between OE lines and WT. (**d**) The allocation of biomass in APER. 0.25 mM N, 0.125 mM NH_4_NO_3_; 1 mM N, 0.50 mM NH_4_NO_3_; 5 mM N, 2.50 mM NH_4_NO_3_. Values are means ± standard (SD) (*n* ≥ 6). Asterisks indicate significant differences between WT and two OE lines (line 1, line 2). * *p* < 0.05; ** *p* < 0.01 (*t*-test).

**Figure 6 ijms-21-07043-f006:**
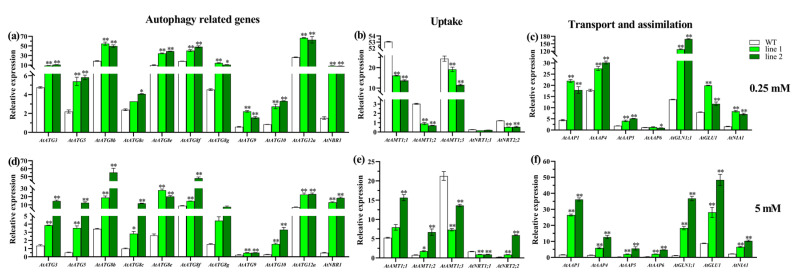
Gene expressions under deficient and sufficient N conditions. The expression of autophagy-related genes in leaves of overexpression *Arabidopsis* (OE) and wild-type (WT) plants under 0.25 mM (**a**) and 5 mM (**d**) N, respectively. The expression of the genes involved in N uptake in roots of OE and WT plants under 0.25 mM (**b**) and 5 mM (**e**) N, respectively. The expression of the genes related to N transport and assimilation in leaves of OE and WT plants under 0.25 mM (**c**) and 5 mM (**f**) N, respectively. 0.25 mM N, 0.125 mM NH_4_NO_3_; 5 mM N, 2.50 mM NH_4_NO_3_. Values are means ± standard (SD) (*n* = 3). Asterisks indicate significant difference between WT and two OE lines (line 1, line 2). * *p* < 0.05; ** *p* < 0.01 (*t*-test). *AAP*, amino acid permease gene; *AMT*, ammonium transporter; *NRT*, nitrate transporter; *NIA*, nitrate reductase encoding gene; *GLU*, Gln 2-oxoglutarate aminotransferase encoding gene; *GLN*, Gln synthetase encoding gene.

**Figure 7 ijms-21-07043-f007:**
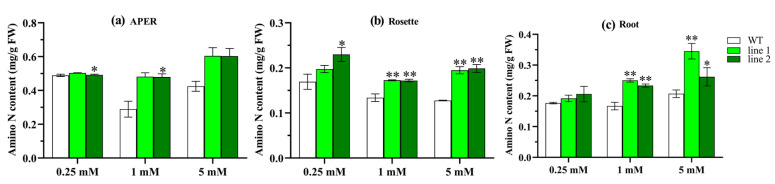
The content of amino N in overexpression *Arabidopsis* (OE) and wild type (WT). (**a**) The amino N content in aerial part excluding rosette (APER). (**b**) The amino N content in rosettes (Rosette). (**c**) The amino N content in roots (Root). 0.25 mM N, 0.125 mM NH_4_NO_3_; 1 mM N, 0.50 mM NH_4_NO_3_; 5 mM N, 2.50 mM NH_4_NO_3_. Values are means ± standard (SD) (*n* = 3). Asterisks indicate significant difference between WT and two OE lines (line 1, line 2). * *p* < 0.05; ** *p* < 0.01 (*t*-test).

**Figure 8 ijms-21-07043-f008:**
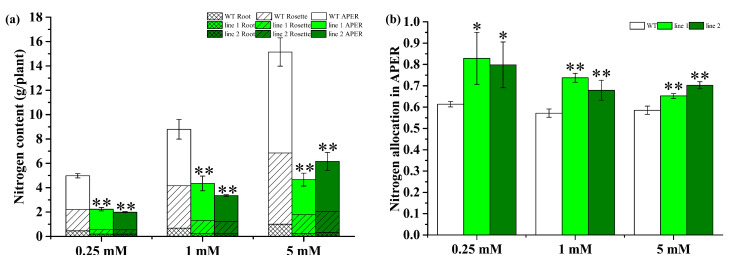
Analysis of N content and allocation in overexpression *Arabidopsis* (OE) and wild type (WT). (**a**) The N composition in OE lines and WT. (**b**) The N allocation in aerial part excluding rosette (APER). 0.25 mM N, 0.125 mM NH_4_NO_3_; 1 mM N, 0.50 mM NH_4_NO_3_; 5 mM N, 2.50 mM NH_4_NO_3_. Values are means ± standard (SD) (*n* = 4). Asterisks indicate significant difference between WT and two OE lines (line 1, line 2). * *p <* 0.05; ** *p <* 0.01 (*t*-test).

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
