# Peer review of "Genome-Wide Identification of CsATGs in Tea Plant and the Involvement of CsATG8e in Nitrogen Utilization"

_ijms, 2020, doi:10.3390/ijms21197043_

Round 1

Reviewer 1 Report

Comments to the Authors,

The submitted manuscript describes the genome-wide identification of CsATGs in Tea Plant and the involvement of CsATG8e in nitrogen utilization. The authors performed set of analyses and experiments in order to identify the Autophagy Related Genes (ATGs) in Camellia sinensis and determined the connection between CsATG8e and nitrogen remobilization. In spite of the fact that the manuscript reports potentially useful data, in its present version can not be accepted for the publication. The major points of criticism can be found below.

  1. The substantial language and style correction, ideally by a native speaker, is essential. I found very numerous language and style errors in the paper, which make it very incomprehensible.

  1. Abstract

Please provide the explanation for “APER” (line 29).

  1. Introduction

The authors should add more information about the macroautophagy including autophagosomes formation. In addition, as core autophagy genes are usually divided into three groups, thus it will be very useful for the readers to introduce this information as well.

Line 55: The authors should provide the information what kind of model plants and cereal crops where used in this studies and what kind of stresses were applied. The adequate references has to be added as well.

Line 57: It should be vegetative phage or phase?

Line 57: Zmatg12 is Zm refers to Zea mays? Please add this information to the text.

Line 72: MdATG8i is Md refers to Malus domestica? Please add this information to the text.

Line 69-72: The authors should add more information about how many of the genes belongs to ATG8 family in different plant species, especially that in this part of the text there is information about overexpression of different ATG8s in different plants without any deeper characterization of this genes.

  1. Results

Line 111: It should be: In addition instead “In Addition”

Line 112: Based on the in silico prediction analysis the authors claim that identified CsATGs localize in many cellular compartments and conclude that this indicates “autophagy related processes occur extensively in tea plant”. Such assumption is highly speculative as the prediction tools do not prove any kind of functional involvement of analysed proteins. The authors should be very careful about the cellular localization of the selected proteins as such analyses should be performed using multiple tools and confirmed experimentally.

Line 193: 12 ATG8 genes were identified in Camellia sinensis, however only the expression of CsATG8e was presented on the Figure 4a. Why the authors decided to analyse only the expression of this particular gene? There is no changes in the expression of another 11 ATG8 genes in the analysed samples? The same question can be applied to the Figure 4b. There is also no information about the standard errors and statistical analysis, which means that there is no information if the observed changes in the expression of selected ATGs genes are significant or not.

Line 225: In Arabidopsis thaliana ATG8 proteins are divided in to three groups based on their amino acid sequences. As the authors compare the amino acid sequences of CsATG8 proteins with AtATG8 sequences it would be very useful to check if CsATG8 proteins can be also divided into specific groups.

Line 232: The authors claim that CsATG8e-GFP protein is localized in nucleus and endoplasmic reticulum, however the green GFP fluorescence can be also visible in the cytoplasm (Image CsATG8e-GFP+N, GFP fluorescence). In general, free ATG8 proteins are localized in the cytoplasm and ATG8-PE proteins are very often used as autophagosome markers. Meanwhile, in the presented Figure there is no CsATG8e-GFP localized to the autophagosomes on the provided images. I also have doubts about the quality of the image CsATG8e-GFP+ER – GFP fluorescence, as the green fluorescence visible on the image is not so bright comparing to the signal observed on the other images. Thus, new images should be provided by the authors in order to prove their assumption that CsATG8e-GFP protein is indeed localized to ER. The authors should also add the information about the excitation and emission spectra used for capturing of the images to exclude the risk that the signals from the GFP and red protein are not overlapping. The CsATG8 localization should also be performed only by using the CsATG8e-GFP construct in order to confirm the nuclear and ER localization without any additional marker proteins.

Line 246: On the Figure 6a we see the image of 15 days old plants, but in the text the authors described 20 day culture, please clarify this.

Line 256: The aerial part except rosette (APER) was use for the analysis of biomass and biomass allocation but based on the images on Figure 6b only rosettes are visible, thus my question is which tissue was used for this experiments? Another question is how do the authors define biomass allocation and how did they measure it?

Lines 259-347: This section is in general vey hard to follow, mainly due to low quality of English and many stylist and grammar errors. I suggest to rewrite it substantially.

In general, the authors very often mix the description of the obtained results with discussion.

  1. Discussion

The discussion in its present form is of poor quality as most of the text concerns the bioinformatics results and barely a short fragment focuses on the autophagy- and nitrogen-related findings.

  1. Material and methods

Lines 446-488: This section of material and methods needs to be reorganized and systematized, especially with respect to used media, as the term “solution with N” is not acceptable. The authors should also give the full name of N- containing salt on each corresponding Figure.

Lines 511-516: There is absolutely no information on the preparation of constructs such as CsATG8e-GFP+N and CsATG8e-GFP+ER. In addition the information on the marker proteins should be given as well as the conditions used for images acquisition.

Author Response

Response to Reviewer 1 Comments

The submitted manuscript describes the genome-wide identification of CsATGs in Tea Plant and the involvement of CsATG8e in nitrogen utilization. The authors performed set of analyses and experiments in order to identify the Autophagy Related Genes (ATGs) in Camellia sinensis and determined the connection between CsATG8e and nitrogen remobilization. In spite of the fact that the manuscript reports potentially useful data, in its present version cannot be accepted for the publication. The major points of criticism can be found below.

Point 1: The substantial language and style correction, ideally by a native speaker, is essential. I found very numerous language and style errors in the paper, which make it very incomprehensible.

Response 1: Thank you for your critical comment. Language and style have been carefully checked throughout the manuscript for clarity and readability, with modifications marked with “Track Changes” in the manuscript. We hope the revised manuscript can reach the standard for publication in the journal.

Point 2: [Abstract] Please provide the explanation for “APER” (line 29).

Response 2: Thank you for pointing this out. We added ‘Aerial part excluding rosette’ after APER (line 28). Full terms are provided for all the abbreviations at their first mention.

Point 3: [Introduction]The authors should add more information about the macroautophagy including autophagosomes formation. In addition, as core autophagy genes are usually divided into three groups, thus it will be very useful for the readers to introduce this information as well. 

Response 3: Thank you for your suggestion. We added information about macro-autophagy in the revised manuscript from line 49 to 52 as you suggested. Additionally, the information about core autophagy genes was supplemented from line 67 to 69.

Point 4: Line 55: The authors should provide the information what kind of model plants and cereal crops where used in this studies and what kind of stresses were applied. The adequate references have to be added as well.

Response 4: Thank you for your advice. The types of plants as well as the treatment conditions applied in this study and related references are shown from line 56 to 63 as you suggested.

Point 5: Line 57: It should be vegetative phage or phase?

Response 5: Thank you for your reminder. It is vegetative phase.

However, while revising the manuscript, we rephrased it as “Wada et al [14] showed that the N remobilization in senescent leaves was suppressed in the mutant Osatg7-1 (Oryza sativa, Os)” (line 57-59).

Point 6: Line 57: Zmatg12 is Zm refers to Zea mays? Please add this information to the text.

Response 6: Thank you for your reminder. We added Zea mays for Zm in line 59.

Point 7: Line 72: MdATG8i is Md refers to Malus domestica? Please add this information to the text. 

Response 7: Thank you for your reminder. We added Malus domestica for Md in line 81. Additionally, full terms were also provided for the following abbreviations: Oryza sativa for Os (line 59), Arabidopsis for At (line 71) and Vitis vinifera for Vv (line 72).

Point 8: Line 69-72: The authors should add more information about how many of the genes belongs to ATG8 family in different plant species, especially that in this part of the text there is information about overexpression of different ATG8s in different plants without any deeper characterization of this genes. 

Response 8: Thank you for your professional suggestion. We added the numbers of genes of the ATG8 family in Arabidopsis, grapevine and rice from line 77 to 79. In addition, experimental conditions were supplemented to characterize these ATG8s from line 80 to 89.

Point 9: [Results]Line 111: It should be: In addition instead of “In Addition”

Response 9: Sorry for the style error. It should be In addition. In the revision, considering the logic of our work, this sentence was removed from our manuscript.

Point 10: Line 112: Based on the in silico prediction analysis the authors claim that identified CsATGs localize in many cellular compartments and conclude that this indicates “autophagy related processes occur extensively in tea plant”. Such assumption is highly speculative as the prediction tools do not prove any kind of functional involvement of analysed proteins. The authors should be very careful about the cellular localization of the selected proteins as such analyses should be performed using multiple tools and confirmed experimentally. 

Response 10: Thank you for your comments. We totally agree with you in that it could be arbitrary and unreliable to predict the cellular localizations without experimental evidence. Therefore, we removed the cellular compartment predictions from our manuscript and replenished the Supplementary Table 1.

Point 11: Line 193: 12 ATG8 genes were identified in Camellia sinensis, however only the expression of CsATG8e was presented on the Figure 4a. Why the authors decided to analyse only the expression of this particular gene? There is no changes in the expression of another 11 ATG8 genes in the analysed samples? The same question can be applied to the Figure 4b. There is also no information about the standard errors and statistical analysis, which means that there is no information if the observed changes in the expression of selected ATGs genes are significant or not. 

Response 11: Thanks for your scientific suggestion. We added the information about the expressions of nine other genes in CsATG8 family in Fuding Dabai, Echa 10 and Zhongcha 108, and updated Figure 4a. As for Figure 4b, we explained the reasons for the selection of the sixteen genes in our experiment from line 220 to 222. We used the lowercase letters in both Figure 4a and Figure 4b to indicate significant difference at P<0.05, with the description for Figure 4a and Figure 4b updated from line 203 to 231.

Point 12: Line 225: In Arabidopsis thaliana ATG8 proteins are divided in to three groups based on their amino acid sequences. As the authors compare the amino acid sequences of CsATG8 proteins with AtATG8 sequences it would be very useful to check if CsATG8 proteins can be also divided into specific groups. 

Response 12: Thank you for your advice. From line 241 to 248, we supplemented the division of ATG8 proteins in Arabidopsis and checked the classification of the CsATG8 proteins.

Point 13: Line 232: The authors claim that CsATG8e-GFP protein is localized in nucleus and endoplasmic reticulum, however the green GFP fluorescence can be also visible in the cytoplasm (Image CsATG8e-GFP+N, GFP fluorescence). In general, free ATG8 proteins are localized in the cytoplasm and ATG8-PE proteins are very often used as autophagosome markers. Meanwhile, in the presented Figure there is no CsATG8e-GFP localized to the autophagosomes on the provided images. I also have doubts about the quality of the image CsATG8e-GFP+ER – GFP fluorescence, as the green fluorescence visible on the image is not so bright comparing to the signal observed on the other images. Thus, new images should be provided by the authors in order to prove their assumption that CsATG8e-GFP protein is indeed localized to ER. The authors should also add the information about the excitation and emission spectra used for capturing of the images to exclude the risk that the signals from the GFP and red protein are not overlapping. The CsATG8 localization should also be performed only by using the CsATG8e-GFP construct in order to confirm the nuclear and ER localization without any additional marker proteins. 

Response 13: Thank you for your suggestion. We agree with you. Indeed, the GFP fluorescence should be visible in the cytoplasm, no CsATG8e-GFP is localized to the autophagosomes, and the image quality needs to be improved. As the current image cannot truly reflect the subcellular localization, we finally removed the original figure 5 and the relevant description in the revised manuscript.

Point 14: Line 246: On the Figure 6a we see the image of 15 days old plants, but in the text the authors described 20 days culture, please clarify this.

Response 14: Sorry for the confusing statement. Actually, the plants in Figure 6a were 20 days old as we mentioned in the text. We revised this in Figure 6a.

Point 15: Line 256: The aerial part except rosette (APER) was use for the analysis of biomass and biomass allocation but based on the images on Figure 6b only rosettes are visible, thus my question is which tissue was used for these experiments? Another question is how do the authors define biomass allocation and how did they measure it?

Response 15: Thank you for your comments. Root, Rosette and APER were used in this experiment, and only Rosette was presented in figure 6b as shown in line 269. For biomass allocation, we defined it as APER / (APER + Rosette) in line 273 to line 274, and measured it by calculating the separate parts as described in line 567.

Point 16: Lines 259-347: This section is in general very hard to follow, mainly due to low quality of English and many stylist and grammar errors. I suggest to rewrite it substantially.

Response 16: Thank you for your critical comment on the confusing statements. In our revised manuscript, we carefully checked the language and style, and rewrote this section as you suggested. The main changes for the language have been listed in Response 1, with the main modifications for this section as follows:

i) We clarified the function of genes we used for RT-qPCR in line 289, line 292 to 303;

ii) In our original manuscript, four paragraphs dealt with the results of gene expression in part 2.4.3. In our revised manuscript, we only presented the changes in the expression of the test genes from line 286 to 303.

iii) In our revised manuscript, we added the full names for some abbreviations in figures from line 284 to 288.

iv) Additionally, the serial numbers for the figures and references have been renewed for this part in our revised manuscript.

Point 17: In general, the authors very often mix the description of the obtained results with discussion.

Response 17: Thank you for your comment. In our revised manuscript, we reorganized the results and deleted some sentences involved in discussion, such as in part 2.4.3 and part 2.4.4 of the original manuscript.

Point 18: [Discussion]The discussion in its present form is of poor quality as most of the text concerns the bioinformatics results and barely a short fragment focuses on the autophagy- and nitrogen-related findings. 

Response 18: Thank you for your comment. We tried to revise the discussion to highlight the relationships between autophagy gene and nitrogen–related indices from line 407 to 463.

Point 19: [Material and methods] Lines 446-488: This section of material and methods needs to be reorganized and systematized, especially with respect to used media, as the term “solution with N” is not acceptable. The authors should also give the full name of N- containing salt on each corresponding Figure.

Response 19: Thank you for pointing this out. This section of materials and methods has been reorganized as well as the media used. According to your suggestion, “solution with N” was replaced “solution containing N” (line 514). Additionally, we have added full name of N- containing salt as ‘NH4NO3in the annotations on Figure 4 to Figure 8 and Supplementary Figure 2 to 3.

Point 20: Lines 511-516: There is absolutely no information on the preparation of constructs such as CsATG8e-GFP+N and CsATG8e-GFP+ER. In addition, the information on the marker proteins should be given as well as the conditions used for images acquisition.

Response 20: Thank you for your suggestion. According to our Response 13, since the experiment of subcellular localization has been removed from the Results part, the corresponding methods were deleted in the revised manuscript.

Reviewer 2 Report

Wei Huang and colleagues presented manuscript about very important Asian crop Camellia sinensis. They showed genome-wide survey in the Camellia genome and identified a total of 80 CsATGs. What is the most important in this study, they proved function of CsATG8e in N remobilization. In my opinion, this manuscript is well written and deserves to be published​.

I have a few questions/suggestions:

In the Introduction Author mentioned that N is taken by the roots, another source is N remobilization from old organs, using autophagy. However, plants may take nitrogen through the leaves - this way should be mentioned by authors (see for example: https://www.nrcresearchpress.com/doi/abs/10.1139/cjfr-2019-0119#.X0VCB8gzbIU   https://www.sciencedirect.com/science/article/pii/S2468014117300079

see also https://www.mdpi.com/2223-7747/9/5/619 ). Any genes connected with foliar N absorption in C. sinensis?

Author used many cultivars of C. sinensis: Fuding Dabai, Fuding Dahao, Huangdan, Tie guanyin, Echa 10, Zhongcha 108, Qianmei 106, Lichuanhong  1, Xianyuzao and Jiukengzao​. Did they find any differences in case results from these cultivars?

Author Response

Response to Reviewer 2 Comments

Wei Huang and colleagues presented manuscript about very important Asian crop Camellia sinensis. They showed genome-wide survey in the Camellia genome and identified a total of 80 CsATGs. What is the most important in this study, they proved function of CsATG8e in N remobilization. In my opinion, this manuscript is well written and deserves to be published.

I have a few questions/suggestions:

Point 1: In the Introduction Author mentioned that N is taken by the roots, another source is N remobilization from old organs, using autophagy. However, plants may take nitrogen through the leaves- this way should be mentioned by authors (see for example: https://www.nrcresearchpress.com/doi/abs/10.1139/cjfr-2019-0119#.X0VCB8gzbIU https://www.sciencedirect.com/science/article/pii/S2468014117300079

see also https://www.mdpi.com/2223-7747/9/5/619). Any genes connected with foliar N absorption in C. sinensis?

Response 1: Thank you for your scientific suggestion. We added the way nitrogen taken through leaves in plants and the relevant references. However, it is a pity that no genes were identified to be responsible for N uptake in C. sinensis. (line 42 to 43).

Point 2: Author used many cultivars of C. sinensis: Fuding Dabai, Fuding Dahao, Huangdan, Tie guanyin, Echa 10, Zhongcha 108, Qianmei 106, Lichuanhong 1, Xianyuzao and Jiukengzao. Did they find any differences in case results from these cultivars?

Response 2: Thank you for your reminding. According to your suggestion, we have simplified our tea cultivars and finally selected three cultivars, including Fuding Dabai and Zhongcha 108, two national cultivars largely grown in China, and Echa 10, a provincial cultivar mainly grown in Hubei province as described from line 504 to 505. In our revised manuscript, with Zhongcha 108 was the only cultivars with unanimous highest expression at S4, the differences among these three tea cultivars are presented in Figure 4a and described from line 213 to 215.

Round 2

Reviewer 1 Report

The manuscript is improved substantially, thus I accept it in present form for publication.